# Adult Patient Risk Stratification Using a Risk Score for Periodontitis

**DOI:** 10.3390/jcm8030307

**Published:** 2019-03-05

**Authors:** Miguel de Araújo Nobre, Ana Ferro, Paulo Maló

**Affiliations:** 1Research and Development Department, Maló Clinic, 1600-042 Lisbon, Portugal; 2Periodontology Department, Maló Clinic, 1600-042 Lisbon, Portugal; aferro@maloclinics.com; 3Oral Surgery Department, Maló Clinic, 1600-042 Lisbon, Portugal; research@maloclinics.com

**Keywords:** periodontitis, teeth, risk, epidemiology

## Abstract

Background: There is a need for analytical tools predicting the risk of periodontitis. The purpose of this study was to estimate and evaluate a risk score for prediction of periodontitis. Materials and methods: This case-cohort study included a random sample of 155 cases (with periodontitis) and 175 controls (randomly sampled from the study population at baseline) that were followed for 3-year. A logistic regression model was used with estimation of the risk ratio (RR) for each potential predictor. Results: The risk model included the predictors “age > 53 years” (RR = 0.53), “smoking” (RR = 2.9), “gingivitis at baseline” (RR = 3.1), “subgingival calculus at baseline” (RR = 1.9), “history of periodontitis” (RR = 2.3), and “less than 2 observations in the first year of follow-up” (RR = 3.7). Patients were distributed into three risk groups based on the preanalysis risk: low risk, moderate risk, and high risk. The risk score discrimination (95% confidence interval (CI)) was 0.75 (0.70; 0.80) (*p* < 0.001, C-statistic). Conclusions: The risk score estimated in the present study enabled to identify patients at higher risk of experiencing periodontitis and may be considered a useful tool for both clinicians and patients.

## 1. Introduction

Periodontitis represents one of the most prevalent disease worldwide. According to the global burden or oral disease report of 2005, the prevalence of periodontitis among adults is between 5 and 20% worldwide [1]. This prevalence estimation was further consolidated in a systematic review and meta-regression, estimating a global age-standardized prevalence of severe periodontitis in the year 2010 at 11.2% (95% uncertainty interval: 10.5–12.0%) [2]. Furthermore, analyzing the average percentages of maximum Community Periodontal Index (CPI) scores among 35–44-year-olds worldwide, indicates that the symptoms of periodontal disease (CPI = 3 or 4) are highly prevalent among adults of all regions, ranging between 37% and 63% [1].

The impact of periodontitis on the quality of life is very high considering the edentulism (affected by chronical oral disease including periodontitis), with a prevalence in senior patients between 26–58% in North America and 13–78% in Europe [1]. Furthermore, regarding the global burden of oral diseases of 2010, periodontal diseases are responsible collectively with dental caries edentulism, oral cancer and cleft lip/palate for 18,814,000 disability-adjusted life years (number of lost years of healthy life), a number that represented an average increase of 45.6% from 1990 to 2010 [3]. The condition is multifactorial, with several variables as potential risk factors and risk indicators, including: previous history of periodontal disease, genetics, bruxism, smoking, diabetes, stress, medication, poor oral health habits, older age, compliance to periodontal monitoring, decreased immunity, systemic diseases, education, poor nutrition, and obesity [4,5,6,7,8,9,10,11,12].

The multifactorial origin of chronical conditions such as periodontitis makes it difficult to accurately diagnose, manage, and predict the risk of a specific patient to develop such disease, representing a challenge for the clinician. Furthermore, it is important to identify the impact of each risk factor, particularly a risk factor with potential to be modified or prevented [13]. Risk algorithms for disease modeling assume an important role in modern Medicine, allowing the clinician to access a tool to aid the diagnostic and decision process [14]. The aim of this study was to estimate and evaluate a risk score for prediction of periodontitis.

## 2. Materials and Methods

The study was approved by an independent Ethical Committee (Ethical Committee for Health, authorization no. 005/2012). This manuscript was written following the guidelines of STROBE (Strengthening the Reporting of Observational Studies in Epidemiology) [15]. This was a prospective case-cohort study based on the data collected from a prospective epidemiological surveillance study on oral diseases [16]. The data was collected from July 2012 to December 2015. An initial cohort of 22,009 participants from the main study [16] was established. From these, there were 19,868 patients with natural teeth (the remaining patients had only implant-supported prostheses and were excluded); with 5355 patients screened in the year 2012 remaining potentially eligible for inclusion considering the required minimum time elapsed for periodontitis to occur. Patients with presence of periodontitis in the first observation (*n* = 430) and patients with periodontally hopeless teeth with planned and executed extraction for full-arch implant rehabilitation (*n* = 91) were excluded from this study. A total of 4834 patients with teeth and without periodontitis in the first observation were eligible for inclusion, from which, 551 registered periodontitis (attack rate of 11.4%) and 4283 registered a healthy dentition during the follow-up of the epidemiological surveillance study (Figure 1). The potential selection bias was addressed by selecting the cases and controls from the same population.

A total of 330 patients were included in this study (*n* = 155 cases; *n* = 175 controls).

One trained outcome assessor was responsible for collecting information from the records.

### 2.1. Sample Size Calculation and Sampling

The sample size calculation was performed using a software program (power and sample size calculations, version 3.0.34, Dupont W.D. and Plummer W.D. Jr., Department of Biostatistics, Vanderbilt University, Nashville, TN, USA). The authors planned a study with of independent cases and controls with 1 control per case. Prior data indicated that the probability of exposure among controls (taking smoking as reference) was 24% [17]. If the true odds ratio for disease in exposed subjects relative to unexposed subjects is 2, we will need to study 155 case patients and 155 control patients to be able to reject the null hypothesis that this odds ratio equals 1 with probability (power) 80%. The Type I error probability associated with this test of this null hypothesis is 5%. Given the nature of the study design (case-cohort study) the number of controls necessary was adjusted.

Considering the number of controls estimated in the previous step (*n* = 155) and the “attack rate” of 11.4%:(1)$$\left(\frac{Patients\;with\;periodontitis}{Total\;patients}\right)⬄(\frac{551}{4834})$$

The sample size equation to adjust the number of controls in a case-cohort study design considers [18]:(2)$$N=number\;of\;controls\;\left(\frac{1}{1-atack\;rate}\right)$$
(3)$$N=155\;\left(\frac{1}{1-0.114}\right)\;⬄\;N=155\;\left(\frac{1}{0.886}\right)\;⬄\;N=155\;\left(1.129\right)\;⬄\;N=175\;\mathrm{controls}$$

A total of 330 patients were selected. The 155 cases (with periodontitis) were selected randomly from the sample of 551 patients with periodontitis. Given the nature of the study design (case-cohort design), the 175 controls were selected randomly from the global sample of 4834 patients (controls could have presence or absence of periodontitis at the end of the study follow-up given the random sample). The random sampling was performed using a random numbers generator (https://www.random.org/).

### 2.2. Variables Definition

The dependent variable “periodontitis” was defined as inflammation of the gingiva and the adjacent attachment apparatus with loss of clinical attachment and loss of adjacent supporting bone loss and according to the American Academy of Periodontology [19]. Gingivitis was defined as inflammation of the gingiva in the absence of clinical attachment loss according to the American Academy of Periodontology [20]. Baseline and follow-up periapical radiographic and clinical evaluations were performed to attest the absence of periodontitis [16]. Concerning the clinical evaluations, the examinations were performed by 22 clinicians that were trained and calibrated to diagnose periodontitis. Training and reliability assessment of dental examinations was conducted within the same day with 30 patients in each annual workshop session, resulting in 90 observations per clinician during the three workshop sessions. The overall interexaminer reliability was estimated using a weighted average of the pairwise interexaminer reliability estimates. The reliability of outcome assessors collecting clinical information assessed through the weighted kappa scores during the three years of follow-up were 0.84, 0.83, and 0.84, respectively.

The independent variables were age (in years and recoded as “≤55 years” or “>55 years”), gender (male or female), socioeconomic status evaluated from the occupation of each patient according to Goldthorpe classification (1: Higher managerial, administrative, and professional occupations; 2: Intermediate occupations; 3: Routine and manual occupations) [21], systemic conditions (presence or absence), diabetes (presence or absence), smoking (smoker, nonsmoker), number of observations in the first year of follow-up for clinical monitoring (<2 observations: 2 or more observations) as a proxy variable for the recall regimen, gingivitis at baseline (presence or absence), subgingival calculus at baseline (presence or absence), type of dentition (only uniradicular teeth, only multiradicular teeth, or both), number of teeth (total number of teeth present in the mouth and recode as loss of up to 8 teeth or loss of more than 8 teeth), dental crowding (presence or absence), dental crowns (presence or absence), bruxism (presence or absence), and history of periodontitis (presence or absence).

### 2.3. Statistical Analysis

#### 2.3.1. Identification of Risk Factors

Descriptive statistics were applied for all variables (measures of central tendency and variance for continuous variables, ratios and frequencies for dichotomous variables). To retrieve the risk model, the statistics were performed according to previously described methods [22]. Univariate logistic regression was performed to all independent variables with estimation of the crude risk ratio (RR) with 95% confidence interval (CI). The independent variables significantly related to the outcome variable at the univariate analysis (*p* < 0.100) were inserted into a multivariable logistic regression model. The variables were introduced in the model without stepwise elimination, and a direct estimate of the adjusted RRs (95% CI) were obtained from the model output [22]. Standard errors of the RR were adjusted through the robust variance estimator method [22].

#### 2.3.2. Development of a Risk Score

The authors developed the risk score based on previously described statistical methods [23,24,25]: the risk score was derived by dividing the beta coefficient of each independent predictor with the base regression coefficient. A sum of weight points for each predictor was calculated to define the final score. In order to compare the incidence of periodontitis, the patients were divided into three risk groups. The cutoff points for the three risk groups were defined based on the multiples of the preanalysis risk: less than half the preanalysis risk (low risk), more than half the preanalysis risk and less than the preanalysis risk (moderate risk), and more than the preanalysis risk (high risk). Robust beta coefficients were calculated for the risk groups were retrieved after bootstrapping based on 1000 bootstrap samples with bias corrected accelerated 95% confidence intervals (95% CI). The risk score discrimination was expressed by the C statistic (95% CI). Statistics were performed using the SPSS version 17 (Statistical Package for Social Sciences, IBM SPSS, New York, NY, USA).

## 3. Results

### 3.1. Participants

The sample of 330 patients included in the study had an average age (standard deviation) of 53.1 years (13.9 years), with a gender distribution of 164 female patients (49.7%) and 166 male patients (51.3%). There were 169 patients with periodontitis (51.2%) and 161 patients without periodontitis (48.8%) (Table 1). The average time of follow-up free from periodontitis (standard deviation) was 19.7 months (11.0 months) for cases and 35.2 months (4.5 months) for controls.

### 3.2. Risk Model

The variables significantly associated with periodontitis and included in the risk model as predictors were “age > 53 years” (RR = 0.53), “smoking” (RR = 2.9), “gingivitis at baseline” (RR = 3.1), “subgingival calculus at baseline” (RR = 1.9), “history of periodontitis” (RR = 2.3), and “less than 2 observations in the first year of follow-up” (RR = 3.7) (Table 2). All the variables included in the model fulfilled the criteria for absence of significant multicollinearity (tolerance > 0.2; variance inflation factor < 2), with the model retrieving high degree of significance (*p* < 0.001 for the goodness of fit).

### 3.3. Risk Score

The algorithm used to determine the predicted probability of periodontitis according to the risk points and risk groups is illustrated in Table 3.

The patients were distributed into three risk groups based on multiples of the preanalysis risk of the data (53.2%): the low risk group considered the patients with less than half the preanalysis risk; the moderate risk group considered the patients with half to less than one times the preanalysis risk; while the high risk group considered the patients with one or more times the preanalysis risk. The observed incidence of periodontitis in the low-, moderate-, and high-risk groups was 23%, 46.5%, and 76.6%, respectively; with a risk score discrimination (95% CI) of 0.75 (0.70; 0.80) (*p* < 0.001, C-statistic) (Figure 2). A clinical situation on the use of the risk score is illustrated (Table 4, Figure 3, Figure 4, Figure 5 and Figure 6).

## 4. Discussion

The present study estimated a risk score for prediction of periodontitis retrieved from a risk algorithm indicating a high degree of significance (*p* < 0.001 for the goodness of fit). Moreover, the model had an acceptable discriminating ability (C-statistic = 0.75) judged by previous publications reporting receiver operating characteristic (ROC) curves with values of the C-statistic between 0.7 and 0.8 [26]. The significance of the present study resided in the development of a risk score for prediction of periodontitis through an epidemiological approach that enabled the simplification of complex statistical calculations, benefiting both clinicians and patients while managing the risk of periodontitis.

One of the limitations of this study include the use of prevalent cases of periodontitis. This limitation was due to the expected extensive time-consuming procedure of studying incident cases (waiting for periodontitis to occur), which, together with the potential susceptibility to sample erosion, would render a highly complicated study design. In an attempt to shorten that limitation, the case-cohort design was chosen to conduct this study. The case-cohort design was attributed to Prentice [27] as an alternative to full cohort design when data collection and follow-up is time-consuming and expensive. The case-cohort design has the particularity of randomly selecting from the source population, regardless of their disease status. Among the advantages of the case-cohort design compared to case-control studies, fact that risk ratios can easily be obtained directly from the cross-product of exposed and unexposed cases and controls, the control group representing a random sample of the source population, and that the control group can easily be used as a reference group to investigate multiple outcomes; while for limitations, a reduced statistical power and the necessity of an increased statistical expertise compared to traditional case-control study designs [18].

Periodontitis is a multifactorial condition that can be influenced by different risk factors in diverse populations, ranging from the host (genetics, host response, and systemic conditions), to environmental (smoking), sociodemographic, oral hygiene habits, compliance to periodontal monitoring, age, nutrition, or obesity [4,5,6,7,8,9,10,11,12].

The six predictors included in the risk score (age, smoking, gingivitis, subgingival calculus, history of periodontitis, less than two observations in the first year of follow-up) were all significant at the multivariable level, and represent predictors that were previously registered as significant risk factors for periodontitis. A significant association was previously described between increased age and periodontitis severity: Page et al. [5] in a longitudinal validation of a risk calculator for periodontal disease registered that despite periodontitis was present in all age groups, the mean bone loss increased from 2.75 mm in patients with 34 years or less of age to 3.75 mm in patients aged between 60 and 74 years. Moreover, a previous history of periodontitis was registered as a risk indicator for periodontitis and the associated prognosis in the same study [5]. Smoking represents probably the major risk factor for periodontitis: a previous study investigating the smoking-attributable periodontitis in the United States from a national survey reported that among current smokers, 74.8% of their periodontitis was attributable to smoking [4]. According to a review [28], one of the potential mechanisms for the effect of tobacco on periodontal health is the stronger inflammatory response with increased release of tissue destructive substances including reactive oxygen species, collagenase, serine proteases, and certain proinflammatory cytokines (such as interleukin-4 and interleukin-13). Gingivitis, subgingival calculus, and less than 2 observations in the first year of follow-up reflect the group of variables related to oral hygiene habits and compliance to periodontal monitoring, all previously accounted as risk factors/indicators for the occurrence of periodontitis [6,29]. Lang and Tonetti [6], in a review of the literature with the objective of establishing a risk model for periodontitis, acknowledged that patients with poor compliance should be considered to be at a higher risk for periodontal disease progression, contributing for a worse prognosis. Zimmermann et al. [29] in a systematic review investigating the effect of tooth brushing frequency on periodontitis, reported a significant overall odds ratio estimate of 1.41 for infrequent compared to frequent tooth brushing in the occurrence of periodontitis. The latter analysis (influence of oral hygiene behaviors) was not performed directly in the present study (however using subgingival calculus and gingivitis as proxy variables), nevertheless it warrants discussion. Inadequate oral hygiene is a known risk factor for periodontitis that could impact both the incidence and treatment. A recent systematic review and meta-analysis of 15 studies investigated the association between oral hygiene and periodontitis, registering significant differences in the effect of oral hygiene on periodontitis (fair versus good oral hygiene: odds ratio = 2.04; poor versus good oral hygiene: odds ratio = 5.01) [30]. However, it is important to point that good oral hygiene habits extend beyond the frequency of tooth brushing: A recent study proposed a new tool for quantifying systematics in tooth brushing behavior, highlighting the importance of both sequence (reaching the sextants and surfaces in a defined order) and time spent brushing (with the best cut-off level at three minutes) [31].

Considering the present study, the predictors included in the risk score represent risk indicators validated in the literature by its use alone or in combination with other factors in previously designed risk models [5,6,10]. A recent systematic review investigated risk factor assessment tools for the prevention of periodontitis progression with the objective of identifying characteristics of patient-based risk assessment tools and reviewing the use of such tools for predicting periodontitis progression [32]. Considering the first objective of the systematic review [32], the authors registered that the majority of risk models consisted in variations of two previous periodontal risk calculators: the periodontal risk calculator [5] and the periodontal risk assessment [6]. Concerning the second objective, the authors concluded that it was possible to predict periodontitis progression and tooth loss using the two risk assessment tools.

Nevertheless, the main difference between the two previously published risk scores [6,10] and the risk score retrieved in the present study resides on the fact that in our study the predictor scores were retrieved directly from the beta coefficients of the multivariable logistic regression model, while in the two previously published risk scores [6,10] the predictor scores were attributed by the authors through a nonstatistical method, thus increasing the probability of sub-optimal estimation of each risk factor weight given the methods’ subjective nature. The third risk score [5] was considered the most complete approach of the three risk scores under the statistical/epidemiological point of view, but was developed under the limitation of including only male patients, rendering predictions of low external validity. External validity is of paramount importance when modeling chronical diseases and was further highlighted by a recent study performed with the objective of validating multivariable models for predicting tooth loss in periodontitis patients [33]. In this validation study, the authors registered low accuracy for most models (with an AUC range between 0.52 and 0.67), concluding that tooth loss prediction in a specific cohort of 301 periodontitis patients was limited [33]. In the present study, the authors used a definition and categorization of risk factors aiming for a high external validity independent of the setting where the evaluation is conducted, nevertheless, the fact that the study was conducted in a private practice advises caution in the generalization of the conclusions for the general population.

Risk scores can be a useful tool for both patients and clinicians. The advantages of a risk score were previously illustrated for another oral condition (peri-implant pathology) [25]: Considering the clinicians, the risk score is an approach for making complex statistical models useful, simplifying the assessment of the multifactorial nature of periodontitis and incorporating it into clinical practice. This represents an effort to provide a tool for clinicians in their decision-making process and to assist them in motivating patients toward healthy behaviors. Regarding the patients, risk scores can be used to induce/motivate behavioral changes in order to reduce the risk score and corresponding periodontitis risk [13]. The use of the risk score for periodontitis over the patients’ follow-up may influence positively the accuracy of periodontal clinical decisions, with a potential impact in the patients’ oral health, reducing both the healthcare cost and the need for complex restorations and/or periodontal therapy.

The limitations of this study include being performed in a single center, the limited follow-up of three years, using prevalent periodontitis cases and the lack of microbiological sampling methods. A further limitation is concerned to the lack of control for oral hygiene habits from the patients that may interfere with the incidence, prevalence, and prognosis of the condition. Longitudinal prospective studies in different populations are needed in the future to test the accuracy of the risk score and further refine the model by including other potential diagnostic tests such as the detection of gene polymorphisms. The study is planned to be continued in order to refine the predictions in a longer term follow-up.

## 5. Conclusions

Within the limitations of this study, it was possible to estimate a risk score for prediction of periodontitis that enabled risk stratification (low risk, moderate risk and high risk) with acceptable discrimination capacity. The risk model included the predictors age > 53 years, smoking, gingivitis at baseline, subgingival calculus at baseline, history of periodontitis, and less than 2 observations in the first year of follow-up. This simple risk score may represent a useful tool for clinicians in the identification, communication and management of periodontitis and also for the patients’ improvement of their self-perceived status of oral health.

## Figures and Tables

**Figure 1 jcm-08-00307-f001:**
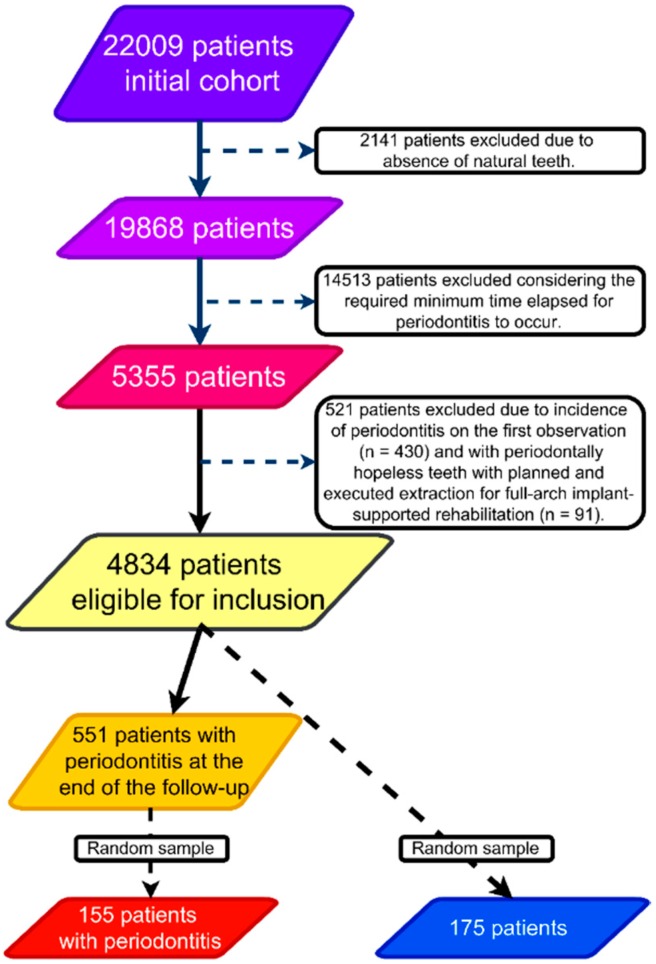
Flowchart illustrative of the case-cohort study design and sampling methods performed in the present study.

**Figure 2 jcm-08-00307-f002:**
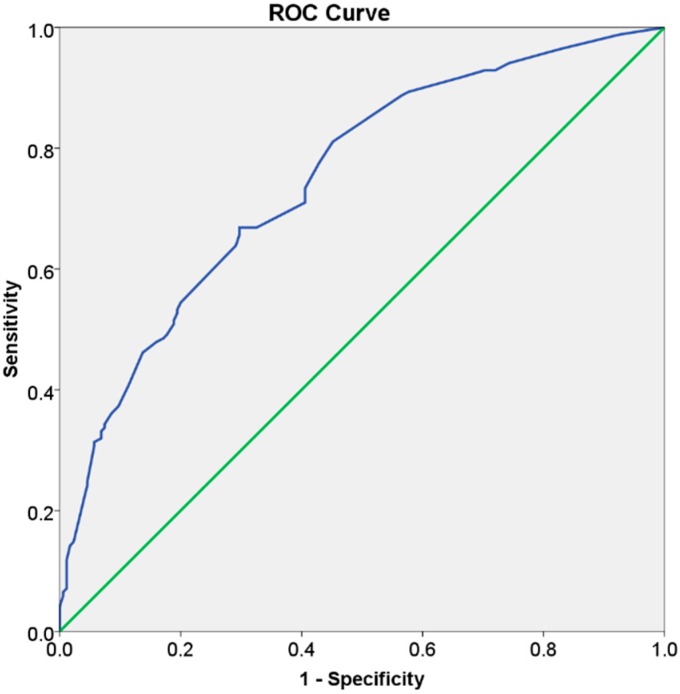
Receiver operating characteristic (ROC) curve illustrating the acceptable discrimination (area under the curve between 0.70 and 0.80) for predicting periodontitis.

**Figure 3 jcm-08-00307-f003:**
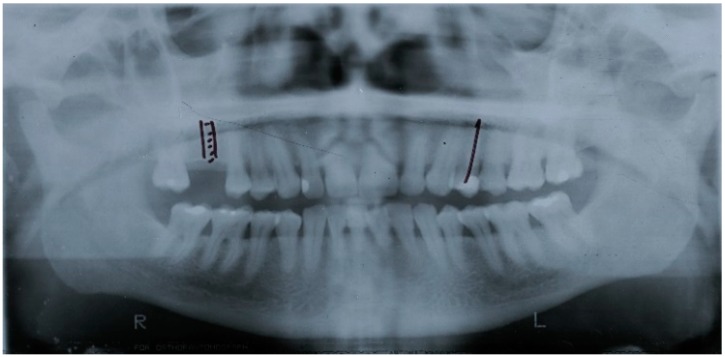
Prediagnosis orthopantomography of a female patient with 70 years of age at baseline.

**Figure 4 jcm-08-00307-f004:**
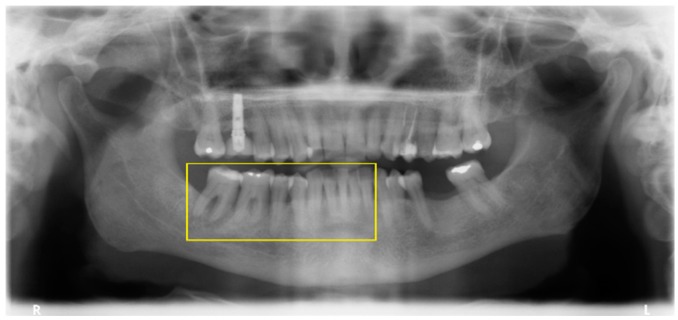
Orthopantomography at diagnosis (two years after baseline) with the patient exhibiting a significant bone loss in the segment marked with a yellow box.

**Figure 5 jcm-08-00307-f005:**
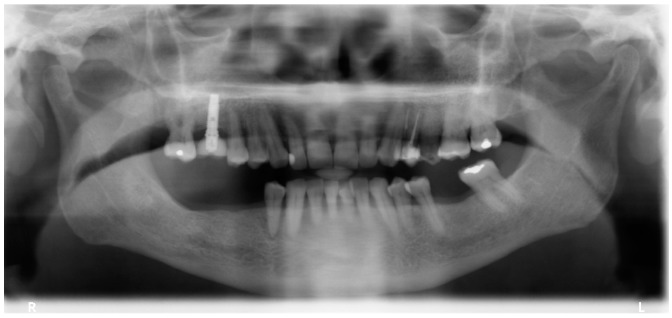
Orthopantomography three years since baseline after extraction of the lower right second premolar, first, and second molars.

**Figure 6 jcm-08-00307-f006:**
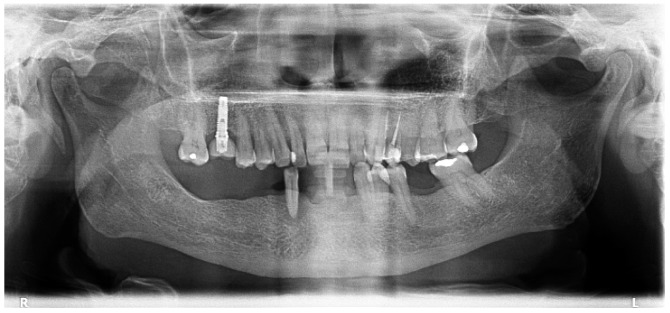
Orthopantomography four years since baseline after extraction of the lower central incisors and right first premolar.

**Table 1 jcm-08-00307-t001:** Sample characteristics and distribution according to cases and controls.

Variables	Total	Cases	Controls
Number (%)	330 (100%)	155 (47%)	175 (53%)
Age mean (standard deviation)	53.1 (13.9)	54.5 (12.9)	51.8 (14.7)
Gender distribution			
Female	164 (49.7%)	76 (46.3%)	88 (53.7%)
Male	166 (51.3%)	79 (47.6%)	87 (52.4%)
Systemic condition number (%)	119(100%)	58 (48.7%)	61 (51.3%)
Hepatitis number (%)	5 (100%)	3 (60%)	2 (40%)
Cardiovascular condition number (%)	62 (100%)	30 (48.4%)	32 (51.6%)
Thyroid number (%)	9 (100%)	6 (66.7%)	3 (33.3%)
Diabetes number (%)	14 (100%)	7 (50%)	7 (50%)
Rheumatologic condition number (%)	17 (100%)	7 (41.2%)	10 (58.8%)
Oncological number (%)	6 (100%)	4 (66.7%)	2 (33.3%)
Inflammatory condition number (%)	2 (100%)	0 (0%)	2 (100%)
Neurologic condition number (%)	1 (100%)	1 (100%)	0 (0%)
Autoimmune condition number (%)	3 (100%)	0 (0%)	3 (100%)
More than one condition number (%)	17 (100%)	9 (52.9%)	8 (47.1%)
Smoking habits number (%)	63 (100%)	40 (63.5%)	23 (36.5%)
1st year number of observations mean (standard deviation)	2.4 (1.0)	2.3 (1.1)	2.6 (0.9)
Gingivitis at baseline (%)	110 (100%)	69 (62.7%)	41 (37.3%)
Socioeconomic Status Code (%)	330 (100%)	155 (47%)	175 (53%)
Level 1 (%)	111 (100%)	56 (50.5%)	55 (49.5%)
Level 2 (%)	77 (100%)	34 (44.2%)	43 (55.8%)
Level 3 (%)	101 (100%)	45 (42.6%)	56 (57.4%)
Nonconsidered level (Pensionaries) (%)	41 (100%)	21 (51.2%)	20 (48.8%)
Presence of periodontitis number (%)	169 (100%)	155 (91.7%)	14 (8.3%)
Type of dentition			
Single-rooted (%)	46 (100%)	44 (95.7%)	2 (4.3%)
Multiradicular (%)	102 (100%)	92 (90.2%)	10 (9.8%)
Both type of teeth (%)	182 (100%)	19 (10.4%)	163 (89.6%)
Presence of subgingival calculus (%)	195 (100%)	105 (53.8%)	90 (46.2%)
Mean number teeth present (standard deviation)	22.5 (7.0)	23.4 (6.4)	21.7 (7.4)
Dental crowding (%)	184 (100%)	89 (48.4%)	95 (51.6%)
Fixed prosthesis supported (%)	44 (100%)	23 (52.3%)	21 (47.7%)
Number patients with bruxism (%)	10 (100%)	4 (40%)	6 (60%)
History of periodontitis (%)	177 (100%)	103 (58.2%)	74 (41.8%)

**Table 2 jcm-08-00307-t002:** Univariable and multivariable risk ratio estimates, multivariable beta coefficients, and risk score points for the prediction of periodontitis.

Variables	Univariable Risk Ratio (95% Confidence Intervals)	Univariable *p*-Value	Multivariable Risk Ratio (95% Confidence Intervals)	Multivariable *p*-Value	Multivariable β Coefficient after Bootstrap Validation (Standard Error)	Risk Score Points
Age						
>53	1.0 (reference)		1.0 (reference)			
≤53	0.67 (0.43; 1.02)	0.062	0.53 (0.31; 0.88)	0.015	−0.64 (0.111)	−1
Gender						
Female	1.0 (reference)					
Male	1.15 (0.76; 1.76)	0.511				
Systemic Problems						
Absent	1.0 (reference)					
Present	0.97 (0.61; 1.54)	0.887				
Diabetes						
Absent	1.0 (reference)					
Present	1.19 (0.42; 3.36)	0.739				
Cardiovascular						
Absent	1.0 (reference)					
Present	0.86 (0.49; 1.48)	0.578				
Rheumatologic						
Absent	1.0 (reference)					
Present	0.47 (0.16; 1.45)	0.190				
More than One Condition						
Absent	1.0 (reference)					
Present	0.65 (0.22; 1.91)	0.432				
Smoking						
Nonsmoker	1.0 (reference)		1.0 (reference)			
Smoker	2.40 (1.38; 4.18)	0.002	2.87 (1.51; 5.42)	0.001	1.05 (0.115)	2
N of observations in the first year						
2 or more	1.0 (reference)		1.0 (reference)			
Less than 2	2.79 (1.51; 5.16)	0.001	3.72 (1.88; 7.34)	<0.001	1.31 (0.115)	2
Gingivitis at baseline						
Absent	1.0 (reference)		1.0 (reference)			
Present	2.43 (1.53; 3.86)	<0.001	3.05 (1.81; 5.14)	<0.001	1.126 (0.104)	2
Dental crowding						
Absent	1.0 (reference)					
Present	1.14 (0.74; 1.74)	0.561				
Subgingival Calculus at Baseline						
Absent	1.0 (reference)		1.0 (reference)			
Present	1.96 (1.26; 3.03)	0.003	1.90 (1.17; 3.10)	0.01	0.64 (0.119)	1
Dental Crowns						
Absent	1.0 (reference)					
Present	1.21 (0.65; 2.28)	0.545				
History of Periodontitis						
Absent	1.0 (reference)		1.0 (reference)			
Present	2.61 (1.69; 4.04)	<0.001	2.26 (1.38; 3.69)	0.001	0.81 (0.124)	1
Type of Dentition		0.207				
Both	1.0 (reference)					
Single-root	4.33 (0.21; 90.85)	0.345				
Multiradicular	4.44 (0.97; 15.88)	0.107				
Number of Teeth Lost						
≤8 teeth	1.0 (reference)					
> 8 teeth	0.75 (0.48; 1.15)	0.184				
Bruxism						
Absent	1.0 (reference)					
Present	0.86 (0.26; 2.87)	0.805				
Socioeconomic Status		0.371				
1st category	1.0 (reference)					
2nd category	0.73 (0.41; 1.28)	0.273				
3rd category	0.71 (0.42; 1.21)	0.204				
*R*^2^ = 0.242; Sensitivity = 52%; Specificity = 73%; Accuracy = 67%						

**Table 3 jcm-08-00307-t003:** Observed incidence of periodontitis in the three risk groups.

Risk Score (Sum of Points)	Risk Group and Predicted Probability Estimated from the Risk Score	Within Group Incidence of Periodontitis	Observed Incidence of Periodontitis
−1–0 points	<26.6%—Low risk (<0.5 preanalysis risk)	14/61 = 23%	14/344 = 4.1%
1–2 points	26.6–53.2%—Moderate risk (0.5 to <1 times preanalysis risk)	74/159 = 46.5%	74/344 = 21.5%
≥3 points	> 53.2%—High risk (>1 times preanalysis risk)	95/124 = 76.6%	95/344 = 27.6%
C-statistic (95% confidence interval) = 0.75 (0.70; 0.80); *p* < 0.001			

**Table 4 jcm-08-00307-t004:** Patient profiling using the risk score at the baseline appointment. According to the clinical evaluation, the presence of gingivitis, subgingival calculus and history of periodontitis were noted, scoring an overall 4 points for a high risk of periodontitis.

Variables	Status	Score
Age less than 53 years	No	0
Smoker	No	0
Less than two observations in the first year of follow-up	No	0
Presence of gingivitis	Yes	2
Presence of sub-gingival calculus	Yes	1
History of periodontitis	Yes	1
Total points	4	
Risk profile	High risk	
Probability of periodontitis	>53.2%	

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
