# Peer review of "Adult Patient Risk Stratification Using a Risk Score for Periodontitis"

_jcm, 2019, doi:10.3390/jcm8030307_

Reviewer 1 Report

The manuscript “Adult patient risk stratification using a risk score for periodontitis” shows that shows that periodontitis risk score influence positively the accuracy of periodontal clinical decisions. In general, good oral hygiene is important in achieving the best clinical response when patients were treated periodontitis. None of any oral hygiene behaviors (frequency of tooth brushing, use of fluoride enriched toothpaste, tongue cleaning) were shown in this study. The other issue is that selection bias may occur in this study.

Major comments:

1.How to select case and control subjects to form 551 registered periodontitis and 4283 registered a healthy dentition is not explained clearly, adds the flowchart can help us easier to understand the research design. 

2.The description of the variable such as N of observations in the first year and healthy number are not clear.

3.C-statistic is not described in the section of materials and methods.

4.Figure 2 to Figure 6 does not represent all 155 clinical situations of cases.

5.The number of cardiovascular condition (N=62) in this study is more than that in diabetes cases (N=17). Why does Table 2 only show diabetes item?

6.There are 41.8% of the control group had a history of periodontitis and the case and control groups had different risk factors of periodontitis such as smoking habits number. Selection bias may occur in this study under those conditions.

7.There should be a discussion of the influences of oral hygiene behaviors, what their role is in clinical periodontal treatment. As this was not measured perhaps a reflection on the literature would be enlightening.

Author Response

Reviewer 1

Comments and Suggestions for Authors

The manuscript “Adult patient risk stratification using a risk score for periodontitis” shows that shows that periodontitis risk score influence positively the accuracy of periodontal clinical decisions. In general, good oral hygiene is important in achieving the best clinical response when patients were treated periodontitis. None of any oral hygiene behaviors (frequency of tooth brushing, use of fluoride enriched toothpaste, tongue cleaning) were shown in this study. The other issue is that selection bias may occur in this study.

Response: The authors thank the Reviewer’s comments. It is true that the oral hygiene behaviors were not accounted for in this study. On the pre-study planning the authors attempted to create a risk score that could be as independent as possible from patient reports, as one thing is what the patient reports on the frequency of tooth brushing, etc, another is what is actually performed. The same issue for a toothpaste enriched with fluoride, or chlorhexidine, or triclosan, or cetylpyridinium chloride, with the difference between Efficacy vs. Effectiveness: We know that in controlled conditions (such as a randomized controlled trial) the active principles work (efficacy), but in the real world the effectiveness is lower because it is dependent on patient adherence. A similar issue for smoking habits, that the authors preferred to treat the variable as dichotomous (smoker, non-smoker) than to introduce the amount of smoking as there is a demonstrated high probability that the patients report values that are not true (usually underestimated). The authors nevertheless agree with the Reviewer and therefore inserted the lack of oral hygiene habits as a limitation on the Discussion section. The selection bias point was treated in the response to review 6.

Changes: Discussion section, 241-251

Major comments:

1.How to select case and control subjects to form 551 registered periodontitis and 4283 registered a healthy dentition is not explained clearly, adds the flowchart can help us easier to understand the research design. 

Response: The authors thank the Reviewer’s suggestion. The Reviewer is correct, the explanation is not clear. A clear explanation of the selection together with a flow chart were introduced as requested.

Changes: Materials and Methods section, lines 59-64, Figure 1, lines 97-97

2.The description of the variable such as N of observations in the first year and healthy number are not clear.

Response: The authors thank the Reviewer’s query. The N of observations were used as a proxy measure for the status of recall and patient adherence and represented clinical appointments (the authors introduced the information in the manuscript); the healthy number was a measure of complete healthy patients that was supplementary and for clarity was erased.

Changes: Results section, Table 1, line 154.

3.C-statistic is not described in the section of materials and methods.

Response: The authors thank the Reviewer’s indication. Please note that the C-statistic was already described in the Materials and Methods section on lines 144 and 145.

Changes: None.

4.Figure 2 to Figure 6 does not represent all 155 clinical situations of cases.

Response: The authors thank the Reviewer’s comment. We assume the Reviewer is referring to the evaluation of clinical characteristics according to Table 1. However, the patient presents 5 of the 6 risk indicators only not qualifying as a smoker (age for cases needed to be > 53 years to qualify as risk, otherwise it represents protective effect). It would be virtually impossible to represent all 155 clinical situations of cases in one subject as there are 64 different combinations of points possible (2 levels of response6 variables). Moreover, if the case is evaluated according to Tables 2 and 3 (according to the number of points), the clinical case is a good representation as it represents the median of points scored (4 points in a scale between -1 to 8, Table 2) and within the group with a higher number of points (> 3 points, high risk; with higher incidence of Periodontitis observed in the sample: 27.6% observed incidence of Periodontitis; Table 3). We remain available to further discuss if the points answered were not what the Reviewer was referring to.

Changes: None.

5.The number of cardiovascular condition (N=62) in this study is more than that in diabetes cases (N=17). Why does Table 2 only show diabetes item?

Response: The authors thank the Reviewer’s query. The presence of diabetes was due to the potential link between diabetes and Periodontitis (on a potential causal relation) easier to disclose than with cardiovascular disease. Nevertheless, the authors introduced the analysis for cardiovascular, rheumatologic and more than one condition (all conditions with at least n=10 cases), without significance on the univariate analysis.

Changes: Results section, Table 2, line 162,163

6.There are 41.8% of the control group had a history of periodontitis and the case and control groups had different risk factors of periodontitis such as smoking habits number. Selection bias may occur in this study under those conditions.

Response: The authors thank the Reviewer’s indication. When planning the study, the sampling strategy and the study design was adapted considering selection bias and therefore a case-cohort study design was selected. The case cohort study design has the advantage (among others) of reducing selection bias as cases and non-cases are selected from the same population (as it was the case of this study where the cases and non-cases were selected from an epidemiological surveillance study with 22009 patients with the particularity that non-cases were randomly selected and therefore reducing the risk of selection bias). Furthermore, the fact that cases and control patients had different frequencies for the variable smoking was dealt as a determinant and not as a source of selection bias as: The source population that gave rise to the cases was the same of the controls; the fact that patients with a history of periodontitis remained as controls was due to maintaining a healthy status versus the cases that developed a reincidence of the disease; among the causes that could significantly influence the development of periodontitis, smoking is one of them, but sub-gingival calculus was another, and a gingivitis state at baseline a third potential cause, all of them with a higher frequency in cases; all of them satisfying a significant number of criteria to be considered causal (Fedak et al. 2015): strength of association, consistency, specificity, temporality, plausibility and experiment.

Fedak KM, Bernal A, Capshaw ZA, Gross S. Applying the Bradford Hill criteria in the 21st century: how data integration has changed causal inference in molecular epidemiology. Emerg Themes Epidemiol. 2015 Sep 30;12:14. doi: 10.1186/s12982-015-0037-4. eCollection 2015.

Changes: None.

7. There should be a discussion of the influences of oral hygiene behaviors, what their role is in clinical periodontal treatment. As this was not measured perhaps a reflection on the literature would be enlightening. 

Response: The authors thank the Reviewer’s suggestion. A discussion of the influences of oral hygiene behaviors and their role in periodontal treatment were introduced.

Lertpimonchai, A.; Rattanasiri, S.; Arj-Ong Vallibhakara, S.; Attia, J.; Thakkinstian, A. The association between oral hygiene and periodontitis: a systematic review and meta-analysis. Int. Dent. J. 2017, 67, 332-343. doi:10.1111/idj.12317. Epub 2017 Jun 23.

Schlueter, N.; Winterfeld, K.; Quera, V.; Winterfeld, T.; Ganss, C. Toothbrushing Systematics Index (TSI) - A new tool for quantifying systematics in toothbrushing behaviour. PLoS. One. 2018, 13, e0196497. doi:10.1371/journal.pone.0196497.

Changes: Discussion section, lines 241-251.

Reviewer 2 Report

Rather questionable but acceptable use of panoramic radiographs for documentation of the alveolar bone loss.

Limited but acceptable time period of the observation. A continuation of the study could be planned?

The assocation between certain cytokine production in smokers and no-smokers could be briefly mentioned in the discussion.

Author Response

Reviewer 2

Comments and Suggestions for Authors

1. Rather questionable but acceptable use of panoramic radiographs for documentation of the alveolar bone loss.

Response: The authors thank the Reviewer’s comment. The panoramic radiographs were only used for the illustration of the clinical case to be able to show the global evolution over time. As the Reviewer pointed correctly, this was not clear and therefore the authors introduced the information that periapical radiographs were used in both baseline and follow-up controls.

Changes: Materials and Methods section, line 93

2. Limited but acceptable time period of the observation. A continuation of the study could be planned?

Response: The authors thank the Reviewer’s comment and suggestion. Yes, the continuation of the study is planned and that information was introduced in the manuscript.

Changes: Discussion section, lines 257,258

3. The association between certain cytokine production in smokers and no-smokers could be briefly mentioned in the discussion.

Response: The authors thank the Reviewer’s suggestion. The discussion was introduced as suggested.

Johannsen A, Susin C, Gustafsson A. Smoking and inflammation: evidence for a synergistic role in chronic disease. Periodontol 2000. 2014;64:111-26. doi: 10.1111/j.1600-0757.2012.00456.x.

Changes: Discussion section, lines 209-213.

Reviewer 3 Report

Review of jcm-441585

This manuscript is attempting to predict chronic periodontitis in adults using risk factors. The authors performed a case-cohort study using both 155 cases and 175 controls according to a previous study (Schouten, et al., Stat. Med. 1993). The model is well considered for statistical analysis, however, the risk factors are ordinary ones. Thus, the factors the authors pointed out by analysis are well-known factors as shown in the abstract. Using these factors, the authors further tried to group the adult patients to risk stratification. The authors showed that this classification fits under 70-80% of AUC in ROC curve. However, the case shown in Figures 2 to 6 seems not to be fully explained by the factors shown by the authors. Clinical experiences may support other stories including occlusal factors caused by missing teeth and prosthodontics including dental implant.

Major concerns;

1. Is the Table 1 for sample characteristics and distribution sufficient to cover all the factors related to the authors' purpose? Others can be found in many publications such as Charalampakis G, et al, Eur J Oral Sci, 2013 and Wu D, et al., PLoS One, 2018.

2. How do the authors apply their predictive factors to the case presented in Figures 2 to 6? Clinical case conference may reach different estimation for the factors involved in the progression of the disease.

3. As the authors discussed in the Discussion section, there are some discrepancies among previous studies and current study, and also the limitation of this study is described, the readers expect a reasonable explanation for the significance of this study.

Minor concerns;

1. Structure of the Tables and percentage shown in them should be re-organized, i.e., location of "Systemic problems" and % for male.

2. Definitions for the clinical evaluations should be described precisely to avoid any misunderstanding.

3.Should follow the STROBE statement.

Author Response

Reviewer 3

Comments and Suggestions for Authors

Review of jcm-441585

This manuscript is attempting to predict chronic periodontitis in adults using risk factors. The authors performed a case-cohort study using both 155 cases and 175 controls according to a previous study (Schouten, et al., Stat. Med. 1993). The model is well considered for statistical analysis, however, the risk factors are ordinary ones. Thus, the factors the authors pointed out by analysis are well-known factors as shown in the abstract. Using these factors, the authors further tried to group the adult patients to risk stratification. The authors showed that this classification fits under 70-80% of AUC in ROC curve. However, the case shown in Figures 2 to 6 seems not to be fully explained by the factors shown by the authors. Clinical experiences may support other stories including occlusal factors caused by missing teeth and prosthodontics including dental implant.

Major concerns;

1. Is the Table 1 for sample characteristics and distribution sufficient to cover all the factors related to the authors' purpose? Others can be found in many publications such as Charalampakis G, et al, Eur J Oral Sci, 2013 and Wu D, et al., PLoS One, 2018.

Response: The authors thank the Reviewer’s query and suggestions. In clinical research, there are always limitations and the present study certainly has them. Considering the comparison with the mentioned studies, Wu et al. investigates possible variables to predict loss to follow-up, including self-efficacy (not included in our study) for the reason that on the pre-study planning the authors attempted to create a risk score that could be as independent as possible from patient reports, as one thing is what the patient reports on the frequency of tooth brushing, etc, another is what is actually performed. Furthermore, clinical attachment loss and probing depth were included as diagnostic criteria and a marker for inflammation, respectively, therefore could not be used as risk indicators for Periodontitis; disease severity could not be used as the aim of the present study was to predict the occurrence of Periodontitis and not its severity, while periodontal surgery as a treatment could not be included to predict the occurrence of the condition. Considering Chralampakis et al., the study is performed on patients with chronic periodontitis at baseline while attempting to identify sites at risk for future progression. From the several factors, the microbiological sampling could be interesting to use on the different context of the present study, attempting to predict the occurrence of disease. Nevertheless, while it is impossible to include all potential variables in one study, the authors included in the Discussion section the discussion on the importance of oral hygiene habits and the non-inclusion of some of the potential risk indicators as limitations.

Changes: Discussion section, lines 241-251, 291-293

2. How do the authors apply their predictive factors to the case presented in Figures 2 to 6? Clinical case conference may reach different estimation for the factors involved in the progression of the disease.

Response: The authors thank the Reviewer’s query. It should be taken in consideration that the case presented in Figures 2 to 6 is purely illustrative. The case presented had the measurement of the risk indicators at baseline with a high risk score, indicative of imminent occurrence of disease. The patient presents 5 of the 6 risk indicators only not qualifying as a smoker (age needed to be > 53 years to qualify as risk, otherwise it represents protective effect). The clinical case is a good representation as it represents the median of points scored (4 points in a scale between -1 to 8, Table 2) and within the group with a higher number of points (> 3 points, high risk; with higher incidence of Periodontitis observed in the sample: 27.6% observed incidence of Periodontitis; Table 3). There are 64 different combinations of points possible (2 levels of response6 variables) and one potential advantage of the risk score is to discriminate and alocate the patient to the risk group in an easy process. Of course that this tool is not intended to replace clinical judgement but rather to support it, for example, by making it easier to communicate with the patient, transmitting responsibility to the patient and working together in the improvement of the risk indicators and therefore improving health gains, a situation that is already reflected on the Discussion section.

Changes: None. 

3. As the authors discussed in the Discussion section, there are some discrepancies among previous studies and current study, and also the limitation of this study is described, the readers expect a reasonable explanation for the significance of this study.

Response: The authors thank the Reviewer’s comment. The significance of the present study resided in the development of a risk score for prediction of periodontitis through an epidemiological approach that enabled the simplification of complex statistical calculations, benefiting both clinicians and patients while managing the risk of periodontitis. The information  was introduced in the manuscript as requested.

Changes: Discussion section, lines 197-200.

Minor concerns;

1. Structure of the Tables and percentage shown in them should be re-organized, i.e., location of "Systemic problems" and % for male.

Response: The authors tank the Reviewer’s indication. The Table was amended as requested.

Changes: Table 1, line 154.

2. Definitions for the clinical evaluations should be described precisely to avoid any misunderstanding.

Response: The authors thank the Reviewer’s indication. The definitions for the clinical evaluations and calibration methods were introduced as requested to avoid any misunderstanding.

Changes: Materials and Methods section, lines 104-111.

3.Should follow the STROBE statement.

Response: The authors thank the Reviewer’s indication. STROBE statement was followed and the information introduced in the manuscript.

Changes: Materials and Methods section, lines 55-56, 60-64, 68-69; SROBE checklist as supplementary file for review.

Reviewer 4 Report

Identifying a panel of factors that are statistical significant in profiling periodontitis  risk for a patient was a task well accomplished by the authors.

I only have a few minor recommendations:

(line 40) - it would be useful for the readers to either extrapolate a bit on the concept of "disability-adjusted-life-years", or to discard the information completely.

(line 54): small typo ("an" instead of "and")

(conclusions): after all this hard work, a larger section for conclusions would seem more appropriate. Please take this into account. 

Author Response

Reviewer 4

Comments and Suggestions for Authors

Identifying a panel of factors that are statistical significant in profiling periodontitis risk for a patient was a task well accomplished by the authors.

I only have a few minor recommendations:

1. (line 40) - it would be useful for the readers to either extrapolate a bit on the concept of "disability-adjusted-life-years", or to discard the information completely.

Response: The authors thank the Reviewer’s suggestion. The authors inserted a short explanation for clarity.

Changes: Introduction section, lines 40,41.

2. (line 54): small typo ("an" instead of "and")

Response: The authors thank the Reviewer’s correction. Proof read and corrected.

Changes: Materials and Methods section, line 54.

3. (Conclusions): after all this hard work, a larger section for conclusions would seem more appropriate. Please take this into account. 

Response: The authors thank the Reviewer’s suggestion. The Conclusions section was improved as suggested.

Changes: Conclusion section, lines 300-303.

Round  2

Reviewer 3 Report

Review of jcm-441585 #2

This manuscript has been modified well according to reviewers' comments. Some parts are well modified, but others still remain, particularly clinical case.

Major concerns;

1. The case presented in Figures 2 to 6 has obviously other factors different from the ones applied by the authors. This reviewer strongly recommend the authors to eliminate this part, and to focus on the statistical analysis and to suggest theoretical suggestions.

2. Is this topic is within the Aim and Scope of this journal? This reviewer await the judge made by the responsible editor(s).